# Spermidine and Rapamycin Reveal Distinct Autophagy Flux Response and Cargo Receptor Clearance Profile

**DOI:** 10.3390/cells10010095

**Published:** 2021-01-07

**Authors:** Sholto de Wet, Andre Du Toit, Ben Loos

**Affiliations:** Department of Physiological Sciences, Stellenbosch University, Stellenbosch 7600, South Africa; 18455468@sun.ac.za (S.d.W.); andredt@sun.ac.za (A.D.T.)

**Keywords:** autophagy, autophagy flux, cargo receptor, co-localisation, recruitment, turnover

## Abstract

Autophagy flux is the rate at which cytoplasmic components are degraded through the entire autophagy pathway and is often measured by monitoring the clearance rate of autophagosomes. The specific means by which autophagy targets specific cargo has recently gained major attention due to the role of autophagy in human pathologies, where specific proteinaceous cargo is insufficiently recruited to the autophagosome compartment, albeit functional autophagy activity. In this context, the dynamic interplay between receptor proteins such as p62/Sequestosome-1 and neighbour of BRCA1 gene 1 (NBR1) has gained attention. However, the extent of receptor protein recruitment and subsequent clearance alongside autophagosomes under different autophagy activities remains unclear. Here, we dissect the concentration-dependent and temporal impact of rapamycin and spermidine exposure on receptor recruitment, clearance and autophagosome turnover over time, employing micropatterning. Our results reveal a distinct autophagy activity response profile, where the extent of autophagosome and receptor co-localisation does not involve the total pool of either entities and does not operate in similar fashion. These results suggest that autophagosome turnover and specific cargo clearance are distinct entities with inherent properties, distinctively contributing towards total functional autophagy activity. These findings are of significance for future studies where disease specific protein aggregates require clearance to preserve cellular proteostasis and viability and highlight the need of discerning and better tuning autophagy machinery activity and cargo clearance.

## 1. Introduction

Macroautophagy is a major intracellular degradation pathway critical in protein removal and the maintenance of cellular homeostasis [1]. Its dysfunction has been associated with the onset of neurodegenerative diseases, typically characterised by the presence of distinct protein inclusion bodies within brain regions associated with the disease [2,3,4]. Indeed, the overall abundance of autophagosomes, the functional unit of autophagy, is increased in neurons during the pathogenesis of Alzheimer’s as well as Parkinson’s disease [5,6]. Moreover, lysosomal storage diseases, such as Gaucher disease [7] or Pompe disease [8] have been characterized by autophagy dysfunction and selective cargo aggregation, with the former presenting a major risk factor for the onset of Parkinson’s disease. Increasing autophagy activity has been shown to aid in the clearance of toxic proteinaceous cargo [9,10,11], therefore, the precision control of autophagy activity and subsequent enhanced removal of particular cargo has been gaining increasing attention [12,13,14,15].

Three variants of autophagy exist, of which macroautophagy is the most characterised and best explored. Macroautophagy, hereafter referred to as autophagy, is primarily activated during starvation conditions and is particularly involved in cytoplasmic bulk degradation targeting mainly long-lived proteins. In doing so, this process generates metabolic substrates and maintains the cell’s energetic state [16,17]. Basal autophagy activity varies according to cell type [18] and serves a “house-keeping” function; eliminating old or damaged cellular components that may lead to the disruption of homeostasis [4]. Furthermore, the expression profile of microtubule-associated protein light chain 3 (LC3), a critical protein recruited to the autophagosome, has been shown to vary substantially between tissue types, supporting the notion of distinct, tissue-specific autophagy activity [18,19]. However, the number of autophagosomes per cell may rise either due to increased autophagosome synthesis, brought about by pharmaceutical induction for example, or due to their accumulation as a consequence of disrupted cellular homeostasis, brought about by proteotoxicity or lysosomal dysfunction [20,21]. Determining the autophagy activity, or autophagy flux, which is defined as the rate at which material is degraded through the entire autophagy pathway [22], has hence received major attention [15,23,24], particularly since increased autophagy activity has been shown to minimise the harmful effects that arise in the pathogenesis associated with neurodegenerative diseases [25,26,27,28]. Autophagy is typically characterised as a sequential process which is initiated by the synthesis of the pre-autophagosome structure; the phagophore, which matures into an autophagosome [29,30]. These will subsequently sequester cytoplasmic components, including proteinaceous cargo, and be delivered to hydrolase-containing lysosomes where degradation takes place [1]. Importantly, recent evidence supports the notion of a careful discernment between the clearance rate of proteins, as autophagy cargo, contrasted by the autophagosome turnover, to better dissect the efficiency and capacity of the cell to clear proteinaceous components [31,32]. Moreover, it is becoming increasingly clear that autophagy is crucial in alleviating cellular stress through the selective degradation of specific cytoplasmic components [33]. In fact, a plethora of complex and dynamically interactive “receptor” proteins exist and engages with the autophagosome machinery, facilitating cargo-specific degradation [34,35,36]. These proteins include domains that allow binding to ubiquitinated cytoplasmic components with LC3-II, thereby enabling the targeting of ubiquitinated proteinaceous cargo to the autophagosome [37]. In this manner, the autophagy system responds selectively by recruiting autophagy receptors and hence, is well equipped to degrade specific cargo targets [13]. A key protein of interest, p62/sequestosome-1, is fundamentally engaged during autophagy progression and is largely used as an additional molecular indicator of autophagy activity, along with LC3-II [26,38,39]. Of note, p62 has been found to be the primary receptor protein involved in aggrephagy, the autophagy-dependent degradation of proteins [34]. Additionally, NBR1 has been described in the degradation of protein aggregates. Indeed, major cross talk between p62 and NBR1 has been revealed, with NBR1 levels increasing upon p62 knockout [35], suggesting a complex interplay between receptor recruitment, cargo clearance and autophagy activity.

Given the crucial role of autophagy activity in controlling neuronal fate through enhanced clearance of particular protein cargo, this relationship deserves urgent attention. The nature of the relationship between autophagosome turnover and receptor clearance is currently largely unclear. It is also unknown to what extent mTOR-dependent or -independent autophagy induction may govern such change in the autophagosome pool, autophagosome turnover and receptor clearance. In light of these complexities, we aimed to examine the relationship between autophagosome turnover and selective receptor recruitment and clearance, focusing on the two predominant cargo receptors, p62 and NBR1. By employing rapamycin [40] and spermidine [41] at low and high concentrations, we set out to explore their concentration- and time-dependent effect on autophagosome activity, receptor recruitment and subsequent clearance by taking advantage of a unique single-cell analysis and micropatterning approach [42,43].

## 2. Methods and Materials

### 2.1. Cell Culture and Maintenance

Mouse embryonic fibroblast (MEF) cells stably expressing green fluorescent protein (GFP)-LC3 (a kind gift from Noboru Mizushima, Tokyo University, Tokyo, Japan) were utilised and cultured in Dulbecco’s modified Eagle’s medium (DMEM, #41965-039, Life Technologies, Johannesburg, South Africa) supplemented with 10% foetal bovine serum (#S-0615, Biochrom, Berlin, Germany), 1% antibiotic-antimycotic (#15240-062, Life Technologies); 100 µg/mL streptomycin, 100 U/mL penicillin, and maintained in a humidified atmosphere with the presence of 5% CO_2_ at 37 °C. MEFs were sub-cultured using trypsin (#25200-072, Life Technologies) to detach adherent cells from flasks. After trypsinisation, DMEM was added in a 2:1 ratio and cells were collected into either a 15 mL (#50015, Biocom Biotech, Centurion, South Africa) or a 50 mL Falcon tube (#50050, Biocom Biotech) depending on the size of the original culture flask. Cells were centrifuged at 1500 rpm for three minutes at room temperature (5804R Centrifuge, Eppendorf, Johannesburg, South Africa). Media was discarded and cells were resuspended in fresh DMEM. Cells were re-seeded into either T25 (#500030, Porvair, Brackenfell, South Africa), T75 (#500029, Porvair) or T175 (#500028, Porvair) culturing flasks or onto in-house fabricated fibronectin patterned coverslips for micropatterning purposes [43].

### 2.2. Treatment Interventions

Rapamycin (Rapa, #R8781, Sigma, Johannesburg, South Africa) and spermidine (Sper, #S2626, Sigma) were employed to pharmacologically induce autophagy. Rapamycin was used at concentrations of 1 µM (H Rapa) and 10 nM (L Rapa) whilst spermidine was used at concentrations of 20 µM (H Sper) and 5 µM (L Sper). Cells were treated for 2, 8 and 24 h, respectively. For autophagy flux assessment, treatment groups were exposed to 400 nM bafilomycin A1 (Baf) for a duration of 2 h following treatment intervention. All drugs were made up using 1 × PBS.

### 2.3. Western Blot Analysis

Culture medium was removed, and cells were rinsed three times with 1 × PBS. Then 200 µL RIPA buffer (Tris-HCl: 10 mM (pH 7.4), 1 mM sodium fluoride (NaF), sodium-deoxycholate (0.1%), SDS (0.1%), 140 mM NaCl, Triton X-100 (1%), 1 mM EDTA, 0.5 mM EGTA, 1 mM PMSF, leupeptin, 1 mM Na_3_VO_4_, aprotonin and benzamidine (1 µg/mL), pepstatin (10 µg/mL)) was added to each flask and cells were subsequently scraped (#500034, Porvair). Next, lysates were collected and sonicated using a MixSonic (S-4000, Fisher Scientific, Johannesburg, South Africa), and then centrifuged at 8000 rpm for 10 min at 4 °C (Spectrafuge 16M, Sigma). The supernatant was collected into 1.5 mL Eppendorf tubes (#P2TUB003C-0001.5, Lasec, Cape Town, South Africa) and stored at −80 °C. 15 µg of protein was mixed with Laemmli’s buffer (1 mL Tis-HCl (0.5 M; pH 6.8), 1.6 mL 10% sodium dodecyl sulphate (SDS, #MKCG7687, Sigma-Aldrich, Johannesburg, South Africa), 0.8 mL glycerol (#1045422, Merck, Johannesburg, South Africa) and 0.4 mL of 0.05% bromophenol blue (#1041675, Merck) in 3.8 mL dH_2_O) in a 2:1 ratio. Samples were subsequently boiled at 95 °C for 5 min. Proteins were separated on a Fast Cast TGX Stain-Free gel (#456-8084, Bio-Rad, Johannesburg, South Africa) consisting of a 12% resolving and 4% stacking component, respectively (1610174, Bio-Rad) and transferred to a PVDF membrane (#170-84156, Bio-Rad) using the Trans-Blot Turbo (#170-4155, Bio-Rad). Membranes were blocked for 2 h using 5% non-fat milk made up in 1x TBS-T (137 mM sodium chloride, 20 mM Tris, 0.1% Tween-20 at pH 7.6). Membranes were washed 3 times for 5 min using TBS-T and incubated with primary anti-bodies overnight at 4 °C. Primary anti-bodies NBR1 (#9891S, Cell Signalling, MA, United States), p62 (#88588, Cell Signalling), β-actin (#4970, Cell Signalling) and LC3B (#2775, Cell Signalling), were used at a 1:5000 dilution in TBS-T, while LAMP2a (#18528, Abcam, Cambridge, United Kingdom) was employed at a 1:1000 dilution. Membranes were washed three times with TBS-T for 5 min and incubated with peroxidase-linked anti-rabbit IgG at a 1:10,000 dilution in TBS-T (#7074S, Cell Signalling) for one hour at room temperature. Finally, membranes were visualised using Clarity Western ECL substrate (#170-5061, Bio-Rad), prepared at a 1:1 ratio and acquired using the ChemiDoc MP System (Bio-Rad).

### 2.4. Micropatterning

Micropatterning was performed to standardise and maintain cell geometry whilst maximising the accuracy of morphometric analysis [44]. In brief, glass coverslips were cleaned using 70% ethanol, rinsed with dH_2_O and left to dry using a drying oven. Next, coverslips were placed into a UVO cleaner (model 16, Jetlight, Irvine, CA, USA) and illuminated for 10 min with deep UV (180 nm) at a distance of 1 cm from the light source. Next, UV exposed slides were placed onto 0.1 mg/mL PLL-g-PEG (Surface Solutions, Zurich, Switzerland) in 10 nM HEPES (pH 7.4). Coverslips were then incubated at room temperature in a light protected manner for 1 h before being washed for 2 min in 1 × PBS and for 2 min in dH_2_O. To achieve the desired disk-shapes, coverslips were placed onto a UV-exposed custom-made photomask (Deltamask, Toppan, Enschede, The Netherlands) bearing the pattern in such a way that the PEG-activated side was in contact with the non-chrome side of the mask. The photomask was then placed into the UVO cleaner with the coverslips facing away from the light source and illuminated for 7 min. Coverslips were detached using dH_2_O and incubated with a solution of 25 µg/mL fibronectin (F1141, Sigma) in 100 mM NaHCO_3_ (pH 8.3) at room temperature for 2 h. Coverslips were then dried at 4 °C and glued onto a custom-made 12-well chamber. Finally, cells were seeded onto micropatterned cover-slips at 1500 cells per well. Media was carefully replaced after 1 h and cells were treated 2 h after being appropriately patterned.

### 2.5. Immunofluorescence and Confocal Microscopy

Following the treatment protocol, cells were fixed using a 1:1 mixture of pre-warmed DMEM and 8% paraformaldehyde. Cells were rinsed 3 times using 1 × PBS and permeabilised using 0.1% Triton X-100 for 6 min. Cells were subsequently washed three times using 1 × PBS, blocked in 3% BSA for 30 min and incubated in either p62 or NBR1 primary antibody at 4 °C overnight. Next, cells were washed with 1 × PBS before adding Alexa Fluor 568 donkey anti-rabbit secondary antibody (#A10037, Life Technologies) for 2 h at 4 °C. Primary and secondary antibodies were made up in a 1:200 dilution using 1 × PBS. Cells were then washed with 1 × PBS and samples mounted using fluorescence mounting medium (Dako, Santa Clara, CA, USA).

### 2.6. Fluorescence Microscopy, Image Acquisition and Analysis

Image acquisition was performed using a Carl Zeiss LSM780 ELYRA PS.1 Super-resolution platform (Carl Zeiss, Oberkochen, Germany) using a LCI Plan Apochromat 63x/1.4 Oil DIC M27 objective and a 405 nm, 488 nm and 633 nm laser line. Raw images were acquired as z-stacks using a minimum of 7 image frames with a step width of 0.40 µm. Micrographs were analysed using Python (Python Software Foundation, version 3.6.3, Beaverton, OR, USA) from the Anaconda1 4.2.0 distribution; Matplotlib (3.1.1), NumPy (1.16.4), OpenCV (3.4.2), Sci-Kit Image (0.21.2) and SciPy (1.3.0.). In brief, image stacks were imported and a Gaussian blur was applied to each frame; the image series was binarized and the foreground segmented with a watershed algorithm; small pixel clusters were removed using a size filter set to a minimum of 5 pixels, and objects were labelled based on the 3D co-localisation of the respective colour channels (Appendix A).

### 2.7. Statistical Analysis

Statistical analysis was performed using GraphPad Prism version 7 (GraphPad Software, San Diego, CA, USA). Two-way analysis of variance (ANOVA) was performed followed by a Fisher’s LSD post-hoc test. Values were represented as the mean ± SEM and significance was accepted at *p* < 0.05.

## 3. Results

Two major pharmaceutical inducers of autophagy have been employed i.e., rapamycin, to induce autophagy in an mTOR-dependent manner [45], and spermidine, which enhances autophagy through deacetylation of EP300 [41]. Western blot analysis was performed to analyse the protein abundance of selected markers at 2 h, 8 h and 24 h post autophagy induction so as to assess initial and subsequent response in autophagy enhancement (Appendix A).

### 3.1. Rapamycin Significantly Increases LC3-II Protein, but Not Receptor Abundance

LC3-II is recruited to phagophores and is a key component of mature autophagosomes. It is therefore a widely used marker to assess their relative abundance. Following H and L Rapa treatments, a significant increase in LC3-II abundance was observed at all given time points compared to the control (Con), except at 24 h following L Rapa exposure both in the presence (+Baf) and absence (−Baf) of Baf [3.22 ± 1.19, 2.56 ± 1.10 (*p* < 0.05)] (Figure 1A). Whilst L Rapa treatment caused a significant increase in LC3-II at 2 h and 8h, this effect was lost at 24 h, suggesting rapid re-establishment of basal autophagy flux after 24 h. Furthermore, following 2 h of H Rapa, there was a significant difference between LC3-II abundance at 2 h +Baf and 2 h −Baf, demonstrating effective autophagy induction. Taken together, a clear concentration-dependent effect of rapamycin on LC3-II abundance levels over time was observed. Indeed, high concentrations of rapamycin appeared to induce a robust increase in autophagy, which was maintained until the 24 h time point. In contrast, a modest increase in autophagy activity following L Rapa treatment was detected, which only lasted for up to 8 h. LAMP2a protein levels responded to H Rapa, especially following 24 h of treatment (Figure 1B). Indeed, LAMP2a abundance significantly increased compared to Con at 24 h −Baf and 24 h +Baf [1.44 ± 0.25, 1.06 ± 0.27 (*p* < 0.05)]. Additionally, LAMP2a protein levels following H Rapa at 24 h −Baf were also shown to be significantly higher than abundance at 2 h −Baf [1.15 ± 0.11 (*p* < 0.05)], and at 24 h +Baf these were significantly higher than at 2 h +Baf [1.60 ± 0.29 (*p* < 0.05)]. These data indicate that H Rapa induces a slow, but gradual, increase in the overall LAMP2a protein abundance. To our surprise, in this model system, p62 and NBR1 protein levels remained largely unchanged. Although p62 abundance was increased in Con +Baf [2.72 ± 0.65 (*p* < 0.05)] compared to the Con, demonstrating effective basal clearance, there was no other condition during which Rapa induced a change in p62 abundance (Figure 1C). Similarly, H Rapa did not cause any change in NBR1 protein levels (Figure 1D). However, an initial increase in NBR1 at 2 h −Baf [1.42 ± 0.09 (*p* < 0.05)] following L Rapa treatment was noted, which returned to basal levels at 8 h.

### 3.2. Spermidine Maintains LC3-II Turnover, whilst Increasing Receptor Protein Levels

Following H Sper treatment LC3-II protein abundance significantly increased compared to Con at 2 h +Baf and 24 h +Baf [8.71 ± 0.29, 3.42 ± 2.03 (*p* < 0.05)] (Figure 2A). Although this does not show effective LC3-II turnover, the significant increase observed may suggest that H Sper does indeed elicit a degree of autophagosome turnover. L Sper on the other hand induced a delayed response in enhanced autophagy activity, as seen by LC3-II protein levels reaching significance at 8 h +Baf [12.26 ± 3.39 (*p* < 0.05)]. However, this increase was relatively short lived as levels returned to basal at 24 h +Baf [4.38 ± 1.37 (*p* < 0.05)]. Taken together, both H Sper and L Sper concentrations elicited a robust autophagosome turnover, which was maintained over 24 h. LAMP2a protein levels remained relatively unchanged following H Sper (Figure 2B). However, LAMP2a levels were significantly reduced at 24 h +Baf [0.68 ± 0.24 (*p* < 0.05)] compared to both Con +Baf and 2 h +Baf [0.86 ± 0.18, 0.89 ± 0.14 (*p* < 0.05)]. To our surprise, a robust response in receptor protein abundance was observed, especially in cells treated with H Sper (Figure 2C). H Sper treatment induced p62 protein levels to rise significantly above Con and 2 h −Baf [1.31 ± 0.11 (*p* < 0.05)] at both 8 h −Baf and 24 h −Baf [3.81 ± 1.14, 4.98 ± 1.75 (*p* < 0.05)] (Figure 2C), indicating a delayed, yet robust increase in the overall p62 abundance that lasted up to, and possibly beyond, 24 h. Of note, this increase in p62 was not observed following L Sper treatment. NBR1 levels followed a similar trend after cells being treated with H Sper (Figure 2D). Here, NBR1 abundance significantly increased compared to Con at 8 h −Baf and 8 h +Baf [2.56 ± 0.32, 2.21 ± 0.33 (*p* < 0.05)] as well as 24 h −Baf and 24 h +Baf [2.62 ± 0.07, 2.73 ± 0.44 (*p* < 0.05)]. Additionally, both 8 h −Baf and 24 h −Baf treatment groups displayed significantly higher NBR1 protein levels than cells at 2 h −Baf [1.27 ± 0.04 (*p* < 0.05)], suggesting a slow increase in NBR1 that reaches significance after 8 h, but not reflective of any flux. Additionally, an initially increased abundance of NBR1 at 2 h −Baf was detected, subsided at 8 h −Baf and 24 h −Baf.

### 3.3. Micropatterning and Confocal Microscopy

The interaction between autophagosomes and their movement towards lysosomes has been previously established [46], including their lysosome fusion profile upon rapamycin and spermidine exposure [43]. Here, we turned our attention to the co-localisation profile between the autophagy machinery and candidate receptors, as point of departure for characterizing their interactions. In order to gain direct insight into occurring changes of autophagy pathway intermediates, their pool sizes and receptor distribution in individual cells, confocal microscopy was performed using micropatterning. This technique enabled the observation and precise quantification of GFP-LC3 puncta, p62 and NBR2 receptor clusters, as well as the co-localisation between either receptor with autophagy machinery.

#### 3.3.1. High Rapamycin Concentration Increases Autophagosomes, Receptors and Their Recruitment, but Not Their Clearance

Following H Rapa, an immediate increase in GFP-LC3 puncta at 2 h was observed, which decreased over time (Figure 3A). Indeed, puncta counts at 2 h and 8 h both in the presence [185.21 ± 20.76, 137 ± 12.28 (*p* < 0.05)] and absence of Baf [125.08 ± 16.39, 101.17 ± 11 (*p* < 0.05)] were significantly higher than at 24 h −Baf and 24 h +Baf [50.08 ± 7.07, 84.63 ± 12.98 (*p* < 0.05)]. Although p62 levels suggested the presence of a turnover under basal autophagy flux conditions, since counts of Con +Baf [88.08 ± 9.50 (*p* < 0.05)] were significantly higher than in Con, there was no enhanced p62 clearance at any other time point for the duration of the experiment. Only cells at 2 h −Baf and 8 h −Baf [115.92 ± 21.59, 88.50 ± 21.63 (*p* < 0.05)] showed p62 puncta significantly higher than the Con (Figure 3B). Surprisingly, the co-localisation analysis between p62 puncta and GFP-LC3 revealed a significant increase at 2 h −Baf [62.33 ± 14.37 (*p* < 0.05)] compared to Con, indicating an increase in the recruitment of p62 to the autophagosome (Figure 3C,F). Furthermore, co-localised puncta at 2 h −Baf were significantly higher than those at 8 h −Baf and 24 h −Baf [29.58 ± 9.14, 20.58 ± 6.07 (*p* < 0.05)], demonstrating a gradual change in the recruitment of p62 over time following H Rapa exposure. Similarly, results revealed the presence of a basal NBR1 turnover as its levels at Con +Baf [168.17 ± 28.34 (*p* < 0.05)] were significantly higher than those at Con (Figure 3D). This pattern was maintained at 2 h −Baf [176.42 ± 21.66 (*p* < 0.05)], albeit NBR1 puncta gradually decreased at 24 h −Baf [8.75 ± 3.31 (*p* < 0.05)] compared to puncta at 2 h −Baf. The GFP-LC3/NBR1 co-localised puncta significantly increased compared to the Con +Baf at both 2 h +Baf and 8 h +Baf [27.08 ± 5.13, 15.83 ± 4.29 (*p* < 0.05)], suggesting effective receptor recruitment (Figure 3E,G). Furthermore, co-localised GFP-LC3/NBR1 puncta at 2 h −Baf were significantly decreased compared to those at 2 h +Baf [27.08 ± 5.13 (*p* < 0.05)]. Indeed, increased NBR1 flux at 2 h −Baf was observed, which was lost after this time point. Of note, the overall total number of NBR1 puncta co-localising with GFP-LC3 was by far smaller than the number of p62 puncta co-localising with GFP-LC3 (Figure 3C), relative to their total cytoplasmic abundance. These results support the notion that overall total p62 and NBR1 receptor availability, their recruitment and subsequent clearance behaviour is largely distinct.

#### 3.3.2. Low Rapamycin Concentration Induces Prolonged Autophagosome Turnover and Effective p62 Clearance

GFP-LC3 puncta counts were significantly increased compared to their +Baf treated counterparts [64.13 ± 9.07 (*p* < 0.05)] at both 2 h and 8 h [126.50 ± 12.42, 143.17 ± 17.03 (*p* < 0.05)] (Figure 4A). Additionally, effective autophagosome turnover was revealed at both 2 h and 8 h timepoints, as demonstrated by a significant difference between 2 h −Baf [83.50 ± 9.42 (*p* < 0.05)] and 2 h +Baf as well as between 8 h −Baf [78.46 ± 14.60 (*p* < 0.05)] and 8 h +Baf, suggesting a prolonged period of enhanced autophagy machinery turnover. It is also worth noting that, unlike in the H Rapa treatment group, GFP-LC3 puncta counts were significantly higher than those in Con at 24 h −Baf [102.79 ± 11.07 (*p* < 0.05)]. Although p62 puncta counts were significantly increased at 2 h −Baf [92.33 ± 15.25 (*p* < 0.05)], an increase in their clearance was only observed at 8 h [60.50 ± 8.90, 143 ± 22.68 (*p* < 0.05)] (Figure 4B). However, increased GFP-LC3/p62 co-localision upon Baf treatment was noted at both 2 h [24.83 ± 9.77 (*p* < 0.05)] and 8 h [24.50 ± 5.49, 70.50 ± 15.06 (*p* < 0.05)] (Figure 4C,F), suggesting robust receptor recruitment at these time points. NBR1 clearance did not markedly increase despite puncta counts being significantly increased at 8 h −Baf and 24 h −Baf [197.92 ± 33.07, 188.75 ± 16.50 (*p* < 0.05)] (Figure 4D,E). Additionally, GFP-LC3/NBR1 co-localisation analysis did not reveal any changes upon Baf treatment, supporting the notion of their slow clearance (Figure 4G).

#### 3.3.3. High Spermidine Concentration Induces Autophagy and Engages Both p62 and NBR1 Recruitment and Clearance

Effective GFP-LC3 flux was observed at both 2 h [81.50 ± 11.73, 157.63 ± 19.16 (*p* < 0.05)] and 8 h [114.79 ± 20.94, 173.21 ± 20.38 (*p* < 0.05)] (Figure 5A), demonstrating a sustained and enhanced autophagosome turnover upon H Sper treatment. p62 puncta counts revealed effective p62 protein clearance as early as 2 h post-treatment [36.50 ± 8.20, 206.58 ± 26.90 (*p* < 0.05)] (Figure 5B). However, this enhanced clearance was lost at 8 h of treatment intervention despite counts of the 8 h −Baf group [111.17 ± 35.41 (*p* < 0.05)] having been significantly higher than in the Con. A return to basal receptor clearance levels was observed at 24 h of treatment [40.75 ± 6.86, 100.42 ± 14.52 (*p* < 0.05)]. GFP-LC3/p62 co-localised puncta mirrored these results (Figure 5C,G). NBR1 clearance was detectable following 2 h [119.50 ± 25.57, 230.92 ± 21.22 (*p* < 0.05)] and 8 h [99.92 ± 20.89, 261.50 ± 45.23 (*p* < 0.05)] of treatment intervention (Figure 5D). However, no tangible change in GFP-LC3/NBR1 co-localisation was observed at 2 h [11.50 ± 4.69, 11.25 ± 1.68 (*p* < 0.05)] (Figure 5E,G), bringing into question the means by which NBR1 is being degraded under these conditions. At 8 h of treatment, a clear increase in GFP-LC3/NBR1 puncta was observed [2.58 ± 0.98, 34.08 ± 12.40 (*p* < 0.05)], suggesting robust recruitment to the autophagosome membrane. This supports previous work by Kirkin et al. [35] demonstrating that NBR1 may act as compensatory receptor upon p62 depletion.

#### 3.3.4. Low Spermidine Concentration Causes Delayed p62 Clearance

Effective autophagosome turnover was detected at 2 h [70.83 ± 8.65, 98.50 ± 9.78 (*p* < 0.05)] and 8 h [69.67 ± 12.57, 110.25 ± 11.40 (*p* < 0.05)] following L Sper treatment (Figure 6A). Additionally, GFP-LC3 puncta counts were significantly decreased at 24 h +Baf [68.71 ± 5.07 (*p* < 0.05)] relative to both 2 h +Baf and 8 h +Baf, suggesting diminishment of the overall autophagosome pool size after 24 h of L Sper treatment. Strikingly, p62 puncta were effectively cleared at 8 h [44.42 ± 6.36, 164 ± 22.87 (*p* < 0.05)] as well as 24 h [61.17 ± 7.44, 108.58 ± 20.15 (*p* < 0.05)], with a greater receptor clearance rate at the 8 h time point (Figure 6B). Additionally, GFP-LC3/p62 co-localisation analysis revealed effective recruitment and clearance at 2 h [30.50 ± 9.46, 57.25 ± 6.16 (*p* < 0.05)] and 8 h [18.67 ± 3.78, 62.58 ± 11.11 (*p* < 0.05)] (Figure 6C,F), supporting the presence of autophagosome turnover rather than the clearance of p62 receptors. However, in both scenarios, it is clear that the extent of p62 clearance was highest at the 8 h time point. In sharp contrast, NBR1 levels did not change significantly, suggesting minor clearance (Figure 6D). However, L Sper treatment gradually increased NBR1 puncta counts, reaching significance at 8 h −Baf [173.33 ± 26.01 (*p* < 0.05)] compared to control levels and the 2 h −Baf group [134.50 ± 23.58 (*p* < 0.05)] at 24 h −Baf [266.92 ± 71.82 (*p* < 0.05)]. Taken together, by using quantitative fluorescence-based single cell analysis, a tremendous increase in both sensitivity and accuracy of the measurements was achieved, evident through the discernment between the total receptor puncta count and their extent of recruitment and co-localisation with GFP-LC3 positive structures.

## 4. Discussion

Autophagy is a crucial role player in the maintenance of cellular homeostasis, in particular proteostasis [47]. The selective nature of autophagy to target proteins to the autophagy machinery through the recruitment of specific receptors makes it a critical biological pathway, particularly given autophagy decline and dysfunction in neurodegeneration [48,49,50]. Although increased autophagic activity has been associated with neuronal protection in a multitude of model systems [14] the relationship between autophagosome turnover, receptor recruitment and protein clearance rate remains poorly understood. Here, we sought to unravel these following autophagy induction through a mTOR-dependent and independent manner using rapamycin [10] and spermidine [41], respectively. Additionally, we sought to expand upon the current knowledge of concentration, and time, -dependent autophagy induction [51,52,53,54] so as to gain better insights into the dynamic nature of the autophagy response.

### 4.1. The Means of Autophagy Induction Impacts Protein Abundance

Our results reveal that treatment with rapamycin and spermidine at different concentrations resulted in unique and dynamic changes in the autophagy system. Both interventions impacted and changed the expression profiles of key autophagy-related proteins in a distinct manner. Firstly, rapamycin induced a significant increase in LC3-II protein abundance greater than the control both in the presence and absence of Baf at multiple time points (Figure 1A), while spermidine appeared to maintain LC3-II protein levels and turnover similar to control levels, especially following 20 µM spermidine treatment (Figure 2A). This was unexpected, as previous work had shown that spermidine treatment induces an increase in LC3-II abundance in various tissue types [53]. Secondly, to our surprise, both p62 and NBR1 protein levels remained largely unchanged following rapamycin treatment, both in the presence and absence of bafilomycin (Figure 1C,D). Thirdly, spermidine treatment at high concentrations led to a gradual increase in both p62 and NBR1 protein abundance, maintaining increased levels at 8 h and 24 h post-treatment (Figure 2C,D). This observation of increased p62 protein levels is contrasted by a previous study where p62 levels diminished upon spermidine treatment [41]. However, it should be noted that U2OS cells had been employed, which will likely exhibit different, cell-inherent flux levels. Indeed, it is known that different tissue- and cell types are characterized by a specific and inherent basal autophagy activity which is regulated according to the respective metabolic demands [14,18]. Moreover, it has been demonstrated that the means of autophagy induction impacts the protein expression profile of the autophagy machinery and autophagy flux dynamics in MEF cells [43].

### 4.2. Autophagosome Turnover Is Functionally Coupled with Receptor Clearance Indicative of Effective Autophagy Flux

Although it is known that both rapamycin and spermidine induce an increase in autophagosome and receptor puncta counts [34,53], how these drugs may change the pool size of autophagosomes and receptor abundance as well as the recruitment profile of receptors to autophagosomes, and hence functionally couple autophagy flux is unknown. Following high concentrations of rapamycin, an immediate increase in the autophagosome count both in the presence and absence of Baf was observed, a response which lasted for 8 h. Moreover, autophagosome pool size was significantly higher at 2 h, demonstrating rapid temporal changes that may be missed when assessing few time points. The subsequent decrease in the autophagosome pool size at 24 h was most likely due to feedback mechanisms following increased amino acid generation, leading to decreased autophagosome synthesis [55]. Importantly, whilst both receptors increased in abundance following treatment, there was no noticeable clearance of proteins following high concentrations of rapamycin, suggesting that the presence and functional removal of receptor and cargo are indeed distinct events. We show that NBR1-associated autophagosomes were cleared at an earlier time point, as autophagosome turnover responded accordingly. This may suggest that NBR1 operates as cargo receptor at a relatively early point in time, indicating that although there was measurable autophagosome turnover, the receptor recruitment appeared to be deficient, thereby leading to decreased clearance of proteinaceous cargo, similar to previous work by Vicente et al., [31]. It is plausible that our high concentrations of rapamycin resulted in a rapid increase and immediate subsequent clearance of p62 and that NBR1 was thus compensating for the lack of p62 protein whilst the p62 pool was being re-established [37,56]. In sharp contrast, low concentrations of rapamycin induced a prolonged, yet effective autophagosome turnover (Figure 4A). Additionally, there appeared to be functional coupling between effective autophagosome turnover, p62 recruitment to the autophagosome and receptor clearance at 8 h, demonstrating the collaboration between these functional entities as they contribute towards increased autophagy flux [39]. These changes are distinct from what was observed following treatment with high concentrations of rapamycin and therefore demonstrate major concentration-dependent effects that govern the autophagy system. In particular, these results highlight that different concentrations of autophagy inducers impact the cargo recruitment and clearance distinctively, which may inform the design of future translational experiments where selective cargo clearance is of importance. Spermidine, a deacetylating agent of EP300, thereby inducing autophagy by increasing the recruitment potential of the Atg12-5-16 complex and LC3 to the phagophore [41], led to a different autophagy response. Following treatment with a high concentration of spermidine, we observed a prolonged autophagosome turnover response at both 2 h and 8 h (Figure 5A). Additionally, p62 and autophagosome-associated p62 puncta clearance was noted at 2 h, followed by a diminished clearance at 8 h. Along with this observation, NBR1 and autophagosome-associated NBR1 puncta clearance rates were elevated at 8 h, supporting the notion of a compensatory role of NBR1 in the absence of p62 [35]. It should be stated however, that p62 levels at 8 h remained significantly elevated compared to the control, hence, an enhanced p62 puncta count remained, however, its involvement in autophagy, as seen by its clearance and co-localisation pattern, was temporarily reduced, and presumably, its role as an aggrephagy receptor has been taken over by NBR1. Indeed, treatment with a high concentration of spermidine was shown to elicit an increased autophagy response distinct from that induced by rapamycin, a direct inhibitor of the mTOR complex. Additionally, high spermidine concentrations maintained an effective autophagy flux response with the involvement of both p62 and NBR1, albeit at differential time points. Acetylation has been shown to play a role in regulating autophagy [41,57], and our use of spermidine may represent some of the changes that arise as a result. The use of alternative means to induce autophagy, such as spermidine, may therefore be favourable as it impacts the compensatory mechanisms of aggrephagy receptors as well as autophagosome synthesis. Furthermore, treating cells with a low or high concentration of spermidine appeared to cause a similar flux response, as autophagosome turnover was effectively engaged at both 2 h and 8 h post-treatment (Figure 6A). Whilst no clearance changes were observed with regards to NBR1 and NBR1-associated autophagosomes, there was an effective clearance of p62-associated autophagosomes (Figure 6C), coupled to autophagosome turnover, further demonstrating the prolonged effect on maintaining autophagy activity. Indeed, autophagosomes, receptors and p62-associated autophagosomes were each characterised by turnover and clearance at 8 h following low spermidine treatment. Interestingly, NBR1 levels as well as NBR1-associated autophagosome levels were shown to be higher at 24 h relative to basal levels, possibly further increasing beyond this point, suggesting that acetylation may affect the regulation of NBR1 synthesis.

### 4.3. Participation of Either Receptor with Autophagosomes Varies and Does Not Manifest Simultaneously

In the present study, both rapamycin and spermidine significantly increased the number of autophagosomes and the abundance of cargo receptors at distinct concentrations. We highlight that, when effective autophagosome turnover is present, there is typically, but not always, effective cargo receptor clearance that follows. However, such effective clearance of receptor-associated autophagosomes engages typically one or the other receptor, not both concomitantly. It should be noted that the number of receptors that co-localised with autophagosomes was far less than expected, given their total cytoplasmic abundance. Indeed, a large reserve pool of non-recruited receptor protein clusters remained in the cytoplasm, demonstrating reserve capacity of the cell to fully engage in receptor recruitment upon favourable stimulation. It may also suggest that substantial alternative receptor functions be held [58]. Indeed, our work shows that induction of autophagy and an increase in autophagosomes does not lead to a complete co-localisation of receptors with autophagosomes, thus maintaining a pool of cytoplasmic receptors available to fulfil alternative cellular functions such as signal integration [59,60,61].

## 5. Conclusions

NBR1 has been shown to play several roles within the cell to maintain cell survival [62,63,64,65]. Additionally, NBR1 is known to play a compensatory role during aggrephagy [37] and our results demonstrate increased co-localisation and clearance of NBR1 given decreased p62-associated autophagosome clearance. However, changing NBR1 levels upon autophagy induction often demonstrates clearance which is not always equally reflected in the clearance of NBR1-associated autophagosomes. It may be the case that NBR1 is, at least in part, also cleared through the endosomal pathway, as supported by Mardakheh et al. [66]. It is known that receptors recruit and deliver cargo to autophagosomes [13]. The ability to discern and align the two processes of cargo recruitment and subsequent degradation with autophagosome turnover, thereby precisely controlling and monitoring the two functional components of autophagy activity will undoubtedly aid in translational research aimed at targeting and clearing toxic protein cargo. Given the distinct localization of the molecular defect in the autophagy pathway in neurodegenerative diseases [14] and the range of autophagy enhancing drugs becoming available, it may be of increasing importance to not only screen [67] and rank [68] autophagy-modulating compounds with respect to autophagic activity, but also to match these with their ability to recruit and clear candidate cargo of interest for therapeutic intervention. Additionally, understanding the extent to which autophagosomes, with co-localised cargo, indeed fuse with and are degraded by lysosomes will become critical in the successful implementation of autophagy flux correction. Therefore, both autophagy machinery turnover and cargo clearance require careful consideration when assessing autophagy function in health and disease.

## Figures and Tables

**Figure 1 cells-10-00095-f001:**
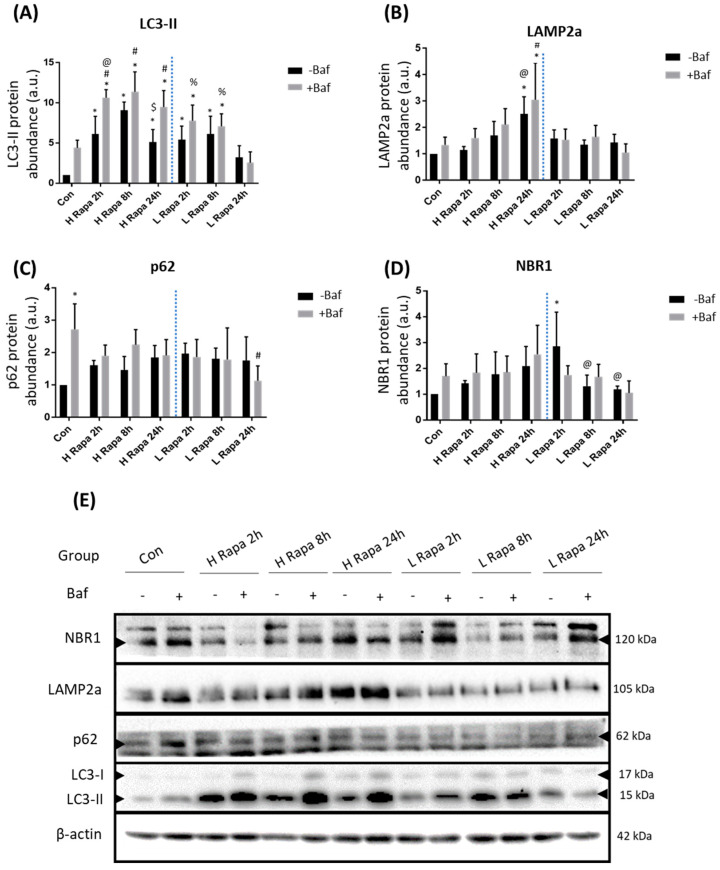
Western blot analysis following 1 µM (H Rapa) and 10 nM (L Rapa) rapamycin incubation over 24 h in the presence (+Baf) and absence (−Baf) of bafilomycin. N = 3. (**A**) LC3-II protein levels. * *p* < 0.05 vs. Con, # *p* < 0.05 vs. Con +Baf, @*p* < 0.05 vs. 2 h −Baf H Rapa, $ *p* < 0.05 vs. 8 h −Baf H Rapa, % *p* < 0.05 vs. 24 h +Baf L Rapa. (**B**) LAMP2a protein levels. **p* < 0.05 vs. Con, #*p* < 0.05 vs. Con +Baf, @*p* < 0.05 vs. 2 h −Baf H Rapa, $ *p* < 0.05 vs. 2 h +Baf H Rapa. (**C**) p62 levels. * *p* < 0.05 vs. Con, #*p* < 0.05 vs. Con +Baf. (**D**) NBR1 protein levels. * *p* < 0.05 vs. Con, @ *p* < 0.05 vs. 2 h −Baf L Rapa. (**E**) Representative western blots are shown.

**Figure 2 cells-10-00095-f002:**
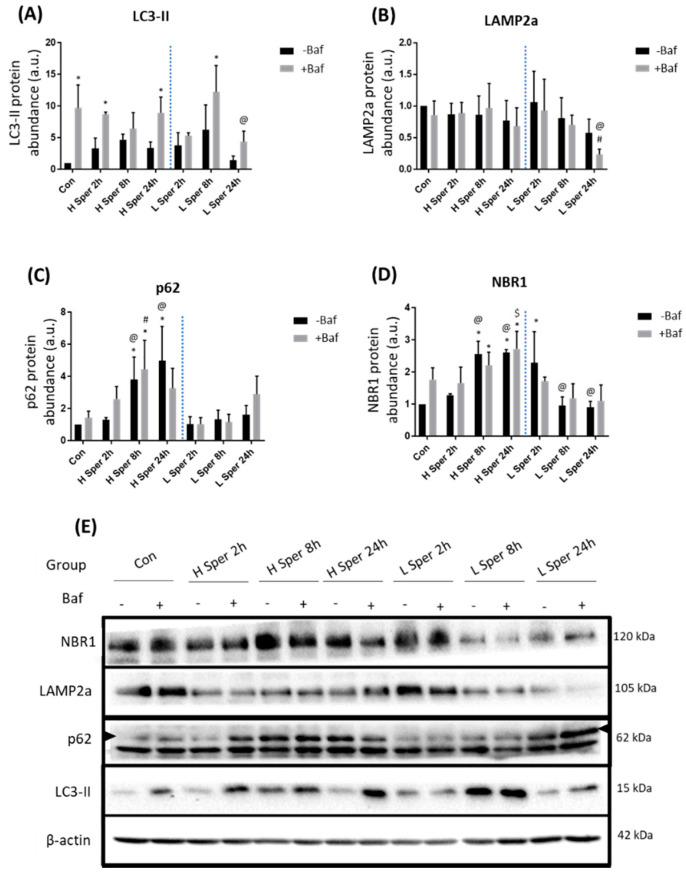
Western blot analysis following 20 µM (H Sper) and 5 µM (L Sper) spermidine incubation over 24 h in the presence (+Baf) and absence (−Baf) of bafilomycin. N = 3. (**A**) LC3-II protein levels. * *p* < 0.05 vs. Con, @ *p* < 0.05 vs. 24 h +Baf L Sper. (**B**) LAMP2a protein levels. # *p* < 0.05 vs. Con +Baf, @ *p* < 0.05 vs. 2 h +Baf L Sper. (**C**) p62 protein levels. * *p* < 0.05 vs. Con, # *p* < 0.05 vs. Con +Baf, @ *p* < 0.05 vs. 2 h −Baf H Sper. (**D**) * *p* < 0.05 vs. Con, @ *p* < 0.05 vs. treatment matched 2 h −Baf, $ *p* < 0.05 vs. 2 h +Baf H Sper. (**E**) Representative western blots are shown.

**Figure 3 cells-10-00095-f003:**
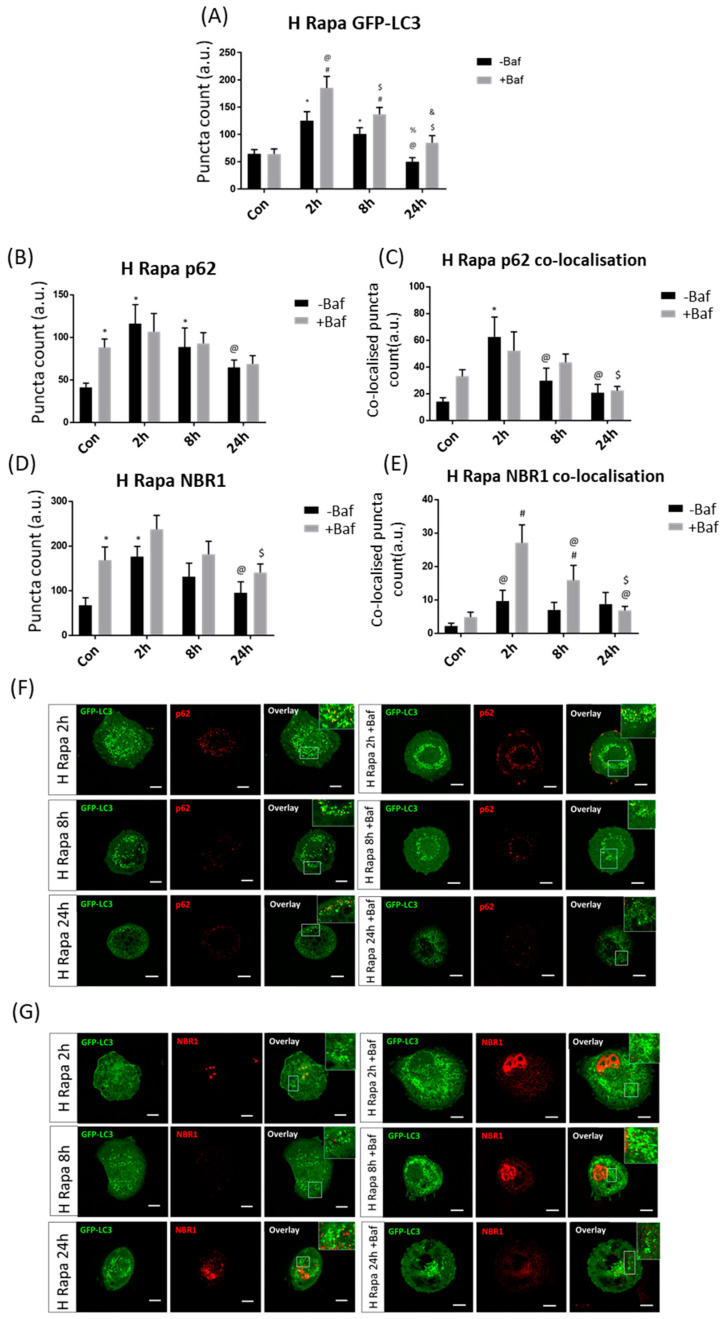
Puncta count analysis following 1 µM rapamycin (H Rapa) over 24 h in the presence (+Baf) and absence (−Baf) of bafilomycin. (**A**) Total GFP-LC3 puncta analysis. N = 24. * *p* < 0.05 vs. Con, # *p* < 0.05 vs. Con +Baf, @ *p* < 0.05 2 h −Baf, $ *p* < 0.05 vs. 2 h +Baf, % *p* < 0.05 vs. 8 h −Baf, & *p* < 0.05 vs. 8 h +Baf. (**B**) Total p62 puncta analysis. N = 12. * *p* < 0.05 vs. Con, @ *p* < 0.05 vs. 2 h −Baf. (**C**) Co-localised GFP-LC3/p62 puncta analysis. N = 12. * *p* < 0.05 vs. Con, @ *p* < 0.05 vs. 2 h −Baf, $ *p* < 0.05 vs. 2 h +Baf. (**D**) Total NBR1 puncta analysis. N = 12. * *p* < 0.05 vs. Con, @ *p* < 0.05 vs. 2 h −Baf, $ *p* < 0.05 vs. 2 h +Baf. (**E**) Co-localised GFP-LC3/NBR1 puncta analysis. # *p* < 0.05 vs. Con +Baf, @ *p* < 0.05 2 h −Baf, $ *p* < 0.05 vs. 8 h +Baf. (**F**) Representative fluorescence micrographs showing GFP-LC3 and p62 puncta. (**G**) Representa-tive fluorescence micrographs showing GFP-LC3 and NBR1 puncta. Scale bar = 10 µm.

**Figure 4 cells-10-00095-f004:**
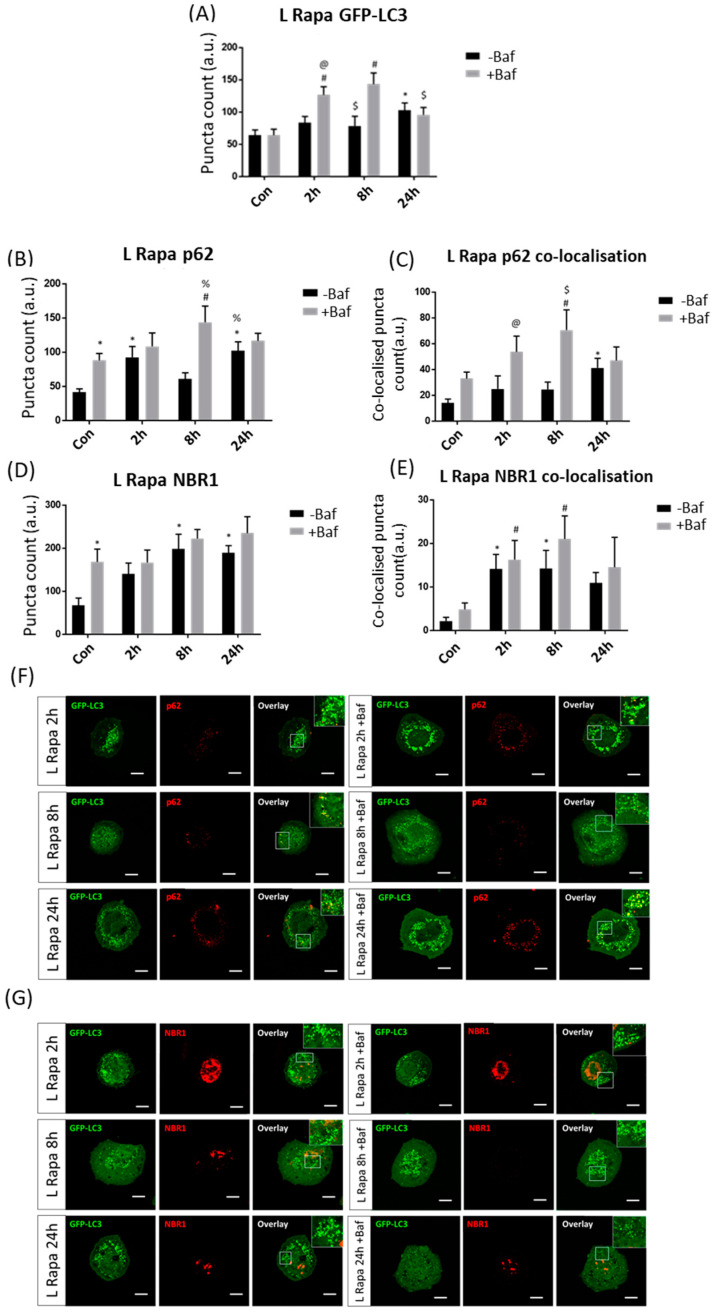
Puncta count analysis following 10 nM rapamycin (L Rapa) over 24 h in the presence (+Baf) and absence (−Baf) of bafilomycin. (**A**) Total GFP-LC3 puncta analysis. N = 24. * *p* < 0.05 vs. Con, # *p* < 0.05 vs. Con +Baf, @ *p* < 0.05 vs. 2 h −Baf, $ *p* < 0.05 vs. 8 h +Baf. (**B**) Total p62 puncta analysis. N = 12. * *p* < 0.05 vs. Con, # *p* < 0.05 vs. Con +, % *p* < 0.05 vs. 8 h −Baf. (**C**) Co-localised GFP-LC3/p62 puncta analysis. N = 12. * *p* < 0.05 vs. Con, # *p* < 0.05 vs. Con +Baf, @ *p* < 0.05 vs. 2 h −Baf, $ *p* < 0.05 vs. 8 h −Baf. (**D**) Total NBR1 puncta analysis. N = 12. * *p* < 0.05 vs. Con. (**E**) Co-localised GFP-LC3/NBR1 puncta analysis. N = 12. * *p* < 0.05 vs. Con, # *p* < 0.05 vs. Con +Baf. (**F**) Representative fluorescence micrographs showing GFP-LC3 and p62 puncta. (**G**) Representa-tive fluorescence micrographs showing GFP-LC3 and NBR1 puncta. Scale bar = 10 µm.

**Figure 5 cells-10-00095-f005:**
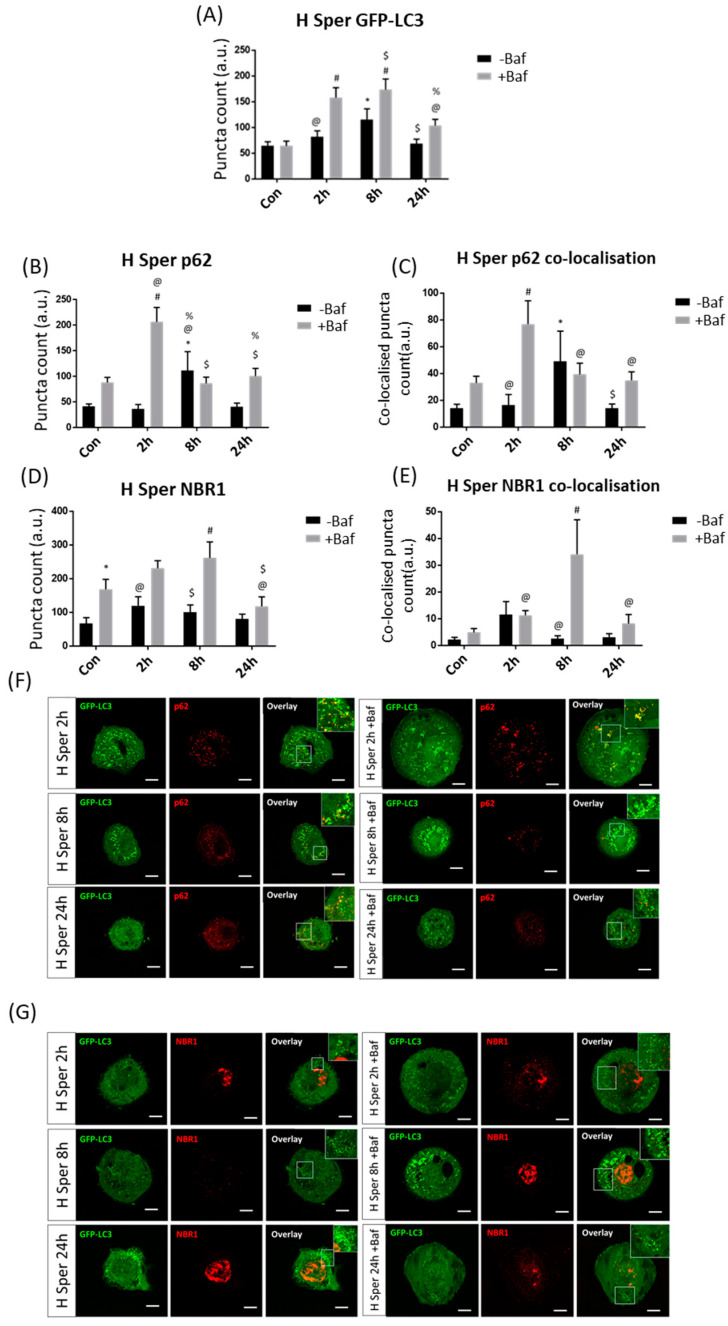
Puncta count analysis following 20 µM spermidine (H Sper) over 24 h in the presence (+Baf) and absence (−Baf) of bafilomycin. (**A**) Total GFP-LC3 puncta analysis. N = 24. * *p* < 0.05 vs. Con, # *p* < 0.05 vs. Con +Baf, @ *p* < 0.05 vs. 2 h +Baf, $ *p* < 0.05 vs. 8 h −Baf, % *p* < 0.05 vs. 8 h +Baf. (**B**) Total p62 puncta analysis. N = 12. * *p* < 0.05 vs. Con, # *p* < 0.05 Con +Baf, @ *p* < 0.05 vs. 2 h −Baf, $ *p* < 0.05 vs. 2 h +Baf, % *p* < 0.05 vs. 24 h −Baf. (**C**) Co-localised GFP-LC3/p62 puncta analysis. N = 12. * *p* < 0.05 vs. Con, # *p* < 0.05 vs. Con +Baf, @ *p* < 0.05 vs. 2 h +Baf, $*p* < 0.05 vs. 8 h −Baf. (**D**) Total NBR1 puncta analysis. N = 12. * *p* < 0.05 vs. Con, # *p* < 0.05 vs. Con +Baf, @ *p* < 0.05 vs. 2 h +Baf, $ *p* < 0.05 vs. 8 h +Baf. (**E**) Co-localised GFP-LC3/NBR1 puncta analysis. N = 12. # *p* < 0.05 vs. Con +Baf, @ *p* < 0.05 vs. 8 h +Baf. (**F**) Representative fluorescence micrographs showing GFP-LC3 and p62 puncta. (**G**) Representa-tive fluorescence micrographs showing GFP-LC3 and NBR1 puncta. Scale bar = 10 µm.

**Figure 6 cells-10-00095-f006:**
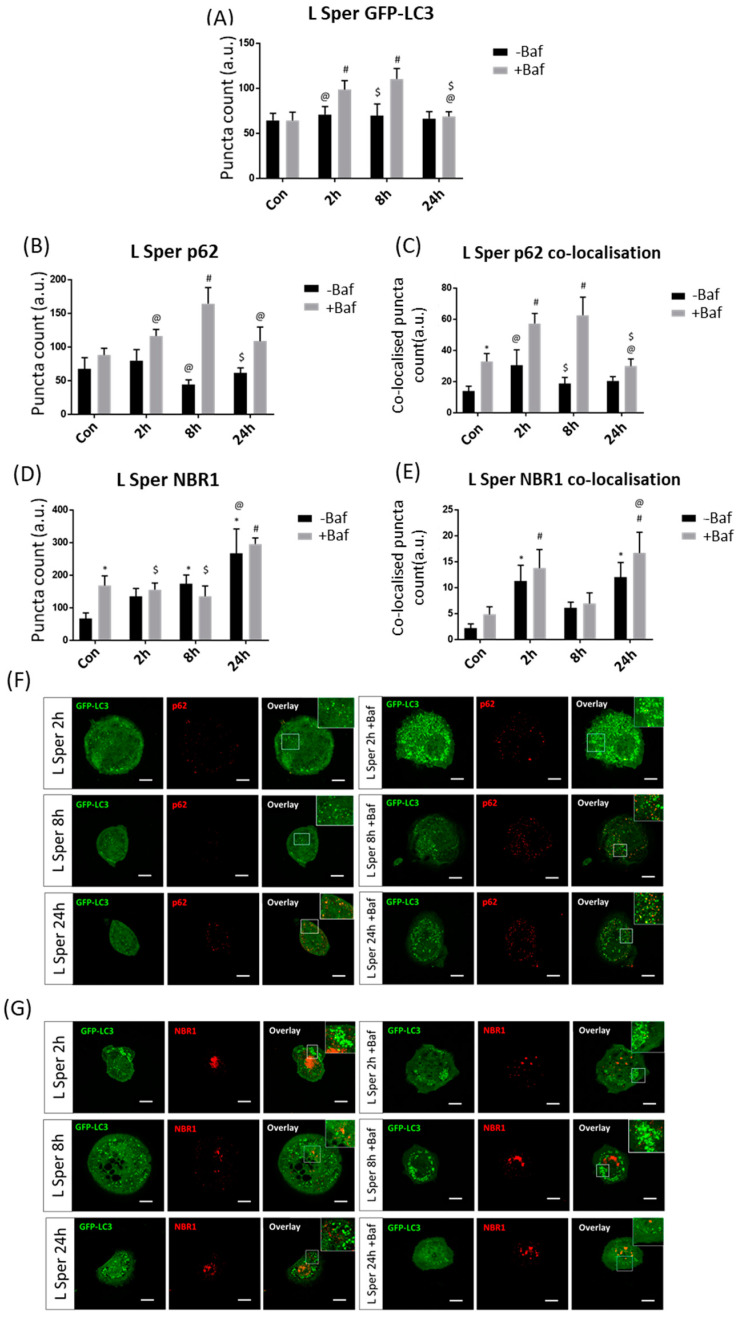
Puncta count analysis following 5 µM spermidine (L Sper) over 24 h in the presence (+Baf) and absence (−Baf) of bafilomycin. (**A**) Total GFP-LC3 puncta analysis. N = 24. # *p* < 0.05 vs. Con +Baf, @ *p* < 0.05 vs. 2 h +Baf, $ *p* < 0.05 vs. 8 h +Baf. (**B**) Total p62 puncta analysis. N = 12. # *p* < 0.05 vs. Con +Baf, @ *p* < 0.05 vs. 8 h +Baf, $ *p* < 0.05 vs. 24 h +Baf. (**C**) Co-localised GFP-LC3/p62 puncta analysis. N = 12. * *p* < 0.05 vs. Con, # *p* < 0.05 vs. Con +Baf, @ *p* < 0.05 vs. 2 h +Baf, $ *p* < 0.05 vs. 8 h +Baf. (**D**) Total NBR1 puncta analysis. N = 12. * *p* < 0.05 vs. Con, # *p* < 0.05 vs. Con +Baf, @ *p* < 0.05 vs. 2 h −Baf, $ *p* < 0.05 vs. 24 h +Baf. (**E**) Co-localised GFP-LC3/NBR1 puncta analysis. N = 12. * *p* < 0.05 vs. Con, # *p* < 0.05 vs. Con +Baf, @ *p* < 0.05 vs. 8 h +Baf. (**F**) Representative fluorescence micrographs showing GFP-LC3 and p62 puncta. (**G**) Representa-tive fluorescence micrographs showing GFP-LC3 and NBR1 puncta. Scale bar = 10 µm.

## Data Availability

The data presented in this study are available on request from the corresponding author. The data are not publicly available due to large, multiple image data files.

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
