# Peer review of "Spermidine and Rapamycin Reveal Distinct Autophagy Flux Response and Cargo Receptor Clearance Profile"

_cells, 2021, doi:10.3390/cells10010095_

Round 1
Reviewer 1 Report
The work is devoted to the relationship between autophagy and segregation of protein material, when a specific protein load is insufficiently recruited into the autophagosome compartment, although the functional activity of autophagy is not weakened. It explores the relationship, interaction of autophagy with receptor proteins such as p62 / Sequestosome16 1 and neighbor of BRCA1 gene 1 (NBR1).
It has been shown that the degree of joint localization of autophagosomes and receptors does not affect the total pool of these components. A high concentration of spermidine induces autophagy and activates the recruitment and clearance of both p62 and NBR1, while rapamycin has little effect on p62 and NBR1 levels. That is, the activation of AMPK kinase by sperminidin does not further trigger signaling along the pathway of mTOR inhibition, characteristic of rapamycin.
Rapamycin and spermidine had different effects on the expression profiles of key autophagy proteins. The exchange of autophagosomes and the selective removal of segregated material are different processes. NBR1 can act as a compensatory receptor for p62 depletion. Removal of disease-specific protein aggregates requires special attention to maintain cellular proteostasis and cell viability.
In general, authors continue their original works, using the interesting and modern approach in study of autophagy flux described earlier:
A Resistive Biosensor for the Detection of LC3 Protein in Autophagy. Viviers C., Du Toit, A., Perold, W., Loos, B., Hofmeyr, J.-H. IEEE Sensors Journal, 2020, 20(10), pp. 5119-5129, 8979366
Loos, B., Klionsky, D.J., Du Toit, A., Hofmeyr, J.-H.S. On the relevance of precision autophagy flux control in vivo–Points of departure for clinical translation. Autophagy, 2020, 16(4), pp. 750-762
The questions:
- In what lysosomal storage diseases (and neurodegeneration), there are changes of autophagy flux (in all?);
- What kind of “therapy” can be suggested for corrections of autophagy flux in these disorders up-today or in perspective.
- 122, 163 Triton-X or Triton X-100?
The work is interesting, presenting the new original results, demonstrating perspective approach to study autophagy flux control (important finally to intralysosomal proteolysis changes) and deserves approval with minimal corrections.
This manuscript can be accepted with small corrections.
Author Response
Ms. No:[Cells] Manuscript ID: cells-1044858
Title:Spermidine and rapamycin reveal distinct autophagy flux response and cargo receptor clearance profile
Authors:Sholto de Wet, Andre du Toit and Ben Loos
Response to reviewers
Response to Reviewer #1:
The work is devoted to the relationship between autophagy and segregation of protein material, when a specific protein load is insufficiently recruited into the autophagosome compartment, although the functional activity of autophagy is not weakened. It explores the relationship, interaction of autophagy with receptor proteins such as p62 / Sequestosome and neighbor of BRCA1 gene 1 (NBR1).
It has been shown that the degree of joint localization of autophagosomes and receptors does not affect the total pool of these components. A high concentration of spermidine induces autophagy and activates the recruitment and clearance of both p62 and NBR1, while rapamycin has little effect on p62 and NBR1 levels. That is, the activation of AMPK kinase by sperminidin does not further trigger signaling along the pathway of mTOR inhibition, characteristic of rapamycin.
Rapamycin and spermidine had different effects on the expression profiles of key autophagy proteins. The exchange of autophagosomes and the selective removal of segregated material are different processes. NBR1 can act as a compensatory receptor for p62 depletion. Removal of disease-specific protein aggregates requires special attention to maintain cellular proteostasis and cell viability.
The work is interesting, presenting the new original results, demonstrating perspective approach to study autophagy flux control (important finally to intralysosomal proteolysis changes) and deserves approval with minimal corrections.
This manuscript can be accepted with small corrections.
We wish to thank the reviewer for the excellent feedback and helpful comments to improve the manuscript further. We have given your recommendations utmost attention and have addressed them comprehensively. Our response to each of the comments is outlined below:
- In what lysosomal storage diseases (and neurodegeneration), there are changes of autophagy flux (in all?)
We thank the reviewer for this excellent comment. We have now included examples of major lysosomal storage disorders, and have also highlighted, where appropriate, the link to enhanced risk for the onset of neurodegeneration (line 37-40). Respective additional references have been included. This addition indeed broadens and enhances the importance of both autophagy flux control and specific cargo clearance.
- What kind of “therapy” can be suggested for corrections of autophagy flux in these disorders up-today or in perspective.
We thank the reviewer for this important comment. We have now included additional information that indicates the wide range of autophagy modulating drugs available, the distinct molecular defect in the autophagy pathway particularly in neurodegeneration, and the importance to screen, rank and match autophagy enhancing drugs according to their ability to induce autophagosome turnover and candidate cargo of interest (line 504-513). Additional references have been included.
- 122, 163 Triton-X or Triton X-100?
We thank the reviewer for locating this error, we have now corrected it throughout the manuscript (indeed, Triton X-100, line 125 and 166)
Reviewer 2 Report
Overall the topic is relevant and timely. Nevertheless, some points need to be addressed to consolidate the conclusions made by the authors.
Reviewer’s concerns about micropatterning and confocal microscopy:
To be able to conclude that the autophagy receptors upon the different tested drugs are or not cleared, authors should perform additional colocalization studies (e.g. colocalization analysis in addition to LC3 between P62 and NBR1 with a lysosomal marker). These analyses should be done for both Rapamycin and Spermidine treatments at the different used concentrations.
Figure 1: blots showing LAMP2 must be improved. This quality of blots could not be used for quantifications. The P62 band could not be cropped as it is (it is the same in Figure 2).
Figure 3 - 6: The legends on the pictures are not readable.
Author Response
Ms. No:[Cells] Manuscript ID: cells-1044858
Title:Spermidine and rapamycin reveal distinct autophagy flux response and cargo receptor clearance profile
Authors: Sholto de Wet, Andre du Toit and Ben Loos
Response to Reviewer #2:
Overall the topic is relevant and timely. Nevertheless, some points need to be addressed to consolidate the conclusions made by the authors.
We wish to thank the reviewer for the helpful comments, allowing us to consolidate the conclusions stronger. We have addressed them as best as possible.
- To be able to conclude that the autophagy receptors upon the different tested drugs are or not cleared, authors should perform additional colocalization studies (g.colocalization analysis in addition to LC3 between P62 and NBR1 with a lysosomal marker). These analyses should be done for both Rapamycin and Spermidine treatments at the different used concentrations.
We thank the reviewer for this important recommendation. We agree with this sentiment and the importance to fully trace specific candidate cargo from recruitment to the autophagosome to the lysosome for subsequent degradation. We have now highlighted the importance of this aspect, indicating the present article as point of departure for such interaction (line 257-261). Additional articles are referred to, where LC3/LAMP2 colocalisation in the context of spermidine and rapamycin has been performed. In addition, we have included a paragraph highlighting future recommendations, to build upon the current work, so as to monitor specific cargo and its targeting to the lysosome (line 509-513). Currently, such work is indeed performed in our laboratory, using model systems of amyloid-and neuronal toxicity, which will form part of an extensive flux/cargo assessment under pathological conditions. We agree that such analyses are indeed of importance to understand the defect and functional capacity of the autophagy pathway.
- Figure 1: blots showing LAMP2 must be improved. This quality of blots could not be used for quantifications. The P62 band could not be cropped as it is (it is the same in Figure 2).
We thank the reviewer for this comment. We have now provided better quality representative western blots for LAMP2 protein and have also corrected the p62 representative western blots respectively (Figure 1 and 2, line 220 and line 250).
- Figure 3 - 6: The legends on the pictures are not readable.
We are especially thankful for this comment since the micrographs are contributing to the core data of this work. We have now adjusted and edited all legends in the micrographs, to enhance readability and visual impact (Figure 3-6, lines 294, 319, 344 and 372 respectively.
Round 2
Reviewer 2 Report
My concerns have been addressed.